# Amphetamine in adolescence induces a sex-specific mesolimbic dopamine phenotype in the adult prefrontal cortex

G. Hernandez [1] ✉, J. Zhao[1], Z. Niu[1], D. MacGowan[2], T. Capolicchio[2], A. Song [1], S. Gul[1], A. Moiz[1], A. Mahmud [1], I. Herrera[1], N. X. Tritsch [1,3], J. J. Day [4] & C. Flores [1,3,5,6] ✉

Drugs of abuse in adolescence impact brain maturation and increase psychiatric risk, with differences in sensitivity between males and females. Amphetamine in early adolescence (postnatal day; PND 21 ± 1–32 ± 1) in male, but not female mice, causes dopamine axons intended to innervate the nucleus accumbens and to grow ectopically to the prefrontal cortex (PFC). This is mediated by drug-induced downregulation of the Netrin-1 receptor DCC. How off-target dopamine axons function in the adult PFC remains to be determined. Here we report that males and females show place preference for amphetamine in early adolescence. However, only in males, amphetamine increases PFC dopamine transporter expression in adulthood (PND 101 ± 15): leading to aberrant baseline dopamine transients, faster dopamine release, and exaggerated responses to acute methylphenidate. Upregulation of DCC in adolescence, using CRISPRa, prevents all these changes. Mesolimbic dopamine axons rerouted to the PFC in adolescence retain anatomical and functional phenotypes of their intended target, rendering males enduringly vulnerable to the harmful effects of drugs of abuse.

In humans, adolescence is a period of extensive biological and behavioral transformation, including profound reorganization of dopaminergic connectivity within the prefrontal cortex (PFC)[1,2]. As a hub for executive function, the PFC matures slowly, showing changes in structure and function well into adulthood[3–6]. As shown in rodent and primate studies, the protracted development of the PFC is accompanied by an increase in the density of dopamine innervation[7–10] which results from dopamine axons still extending from the nucleus accumbens (Nac) to the PFC[11–13]. This delayed growth is the only documented instance of long-distance axonal pathfinding in adolescence[13,14] and depends on the precise signaling of the guidance cue receptor DCC (Deleted in Colorectal Cancer) and its ligand Netrin-1[11,13].

In mice, exposure to recreational-, but not therapeutic-like doses of amphetamine (AMPH) in early adolescence decreases the expression of *Dcc* mRNA[11] and DCC protein[15] in dopamine neurons as well as of Netrin-1 protein levels in the Nac[11,16]. These disruptions cause mistargeting of dopamine axons, leading to the rerouting of mesolimbic projections to the PFC and to a reduced number of dopamine axon varicosities in this region[11,13,17]. These changes trigger alterations in the dendritic architecture of PFC pyramidal neurons, resulting in enduring impairment in inhibitory control[11,18]. Notably, AMPH-induced disruption of adolescent dopamine and cognitive development is sexually dimorphic-it occurs in male mice only[11]. Furthermore, CRISPR activation (CRISPRa)-mediated upregulation of DCC receptors in the ventral tegmental area (VTA) in early adolescence in male mice induces female-like protection against AMPH-induced axon mistargeting and cognitive control dysfunction[11].

Mesolimbic and mesocortical dopamine projections have distinct anatomical and functional characteristics across species[19–22]. Here we assessed if AMPH-induced disruption of dopamine axon pathfinding in adolescence impacts PFC dopamine dynamics in adulthood and if ectopic mesolimbic dopamine axons retain their anatomical and functional phenotype or adopt properties of their off target[19–21]. We addressed these questions by combining behavioral and quantitative neuroanatomical analyses, dopamine dynamics monitoring, and CRISPRa gene editing in male and female mice.

## Methods
### Animals
Experimental procedures were performed according to the guidelines of the Canadian Council of Animal Care and approved by the McGill University/ Douglas Mental Health University Institute Animal Care Committee.

[1]Douglas Mental Health University Institute, Montréal, QC, Canada. [2]Integrated Program in Neuroscience, McGill University, Montréal, QC, Canada. [3]Department of Psychiatry McGill University, Montréal, QC, Canada. [4]Department of Neurobiology, University of Alabama at Birmingham, Birmingham, AL, USA. [5]Department of Neurology and Neurosurgery, McGill University, Montréal, QC, Canada. [6]Ludmer Centre for Neuroinformatics & Mental Health, McGill University, Montréal, QC, Canada. ✉e-mail: giovanni.hernandez.comtl@ssss.gouv.qc.ca; cecilia.flores@mcgill.ca

C57BL/6 mice were obtained from Charles River Laboratories (Saint-Constant, QC, Canada). All mice were maintained on a 12-h light–dark cycle (light on at 0800 h) in a temperature controlled (21 C) facility with 42% humidity and given ad libitum access to food and water. Male and female mice were housed with same-sex littermates throughout the experimental procedures. Mice were assigned randomly to each experimental condition using a computer random number generator. The potential confounding effect of cage location within the animal facility was not controlled.

### Drugs and dose
d-Amphetamine sulfate (Sigma-Aldrich, Dorset, United Kingdom; AMPH) was dissolved in 0.9% saline. All AMPH injections were administered i.p. at a volume of 0.1 mL/10 g. A rewarding dose of AMPH (4 mg/Kg) was used across all the experiments[11,23]. As in all our previous work, we refer to this AMPH regimen as "recreational-like" dose exposure, since this dose achieves peaks plasma levels in adolescent mice comparable to those seen in human recreational use[23].

Methylphenidate (Apotex, Canada) was dissolved in 0.9% saline, and was administered i.p. at a volume of 0.1 mL/10 g. A single dose of 10 mg/kg was selected because it reliably increases extracellular dopamine levels in the PFC[24–26].

### Conditioned place preference
Male and female wild-type mice were separately assessed in a biased conditioned place preference (CPP) paradigm using 4 mg/kg AMPH or saline during early adolescence (PND 21–32), as previously described[11,27]. On PND20, each mouse freely explored the CPP apparatus (two distinct chambers connected by a neutral area) for 20 min. Initial chamber preference was determined by calculating the percentage of time spent in each chamber. The less preferred chamber was then paired with AMPH (experimental group) or saline (control group) in alternating 30-min conditioning sessions over 10 days (5 AMPH, 5 saline). Control mice received saline exclusively. On PND32 and PND80, a 20-min conditioning test was conducted to assess chamber preference. Delta preference score (Δ Place Preference) was calculated as: % time spent in the initially unpreferred chamber (post-test) – % time spent in the same chamber (pre-test).

### Stereotaxic surgeries
**Fiber photometry**. When reaching adulthood (PND81 ± 1) a subset of animals tested in the conditional place preference paradigm were anesthetized with Isoflorane (Fresenius Kabi, Canada) and then placed in a stereotaxic frame for microinfusion of the viral construct using a Stoelting Quintessential Stereotaxic Injector attached to a Hamilton 5 μL microsyringe 75RN. An Adeno-associated virus (AVVs) expressing hSyn-rDA2h (Grab$_{DA2h}$)[28] (Canadian Neurophotonics Platform Viral Vector Core Facility RRID:SCR_016477) was injected (0.5 μL, tier $4 \times 10^{-12}$ vg/mL) unilaterally into the left infralimbic region of the PFC using the following coordinates: AP: +1.8 mm relative to Bregma, ML: 0.3 mm DV - 2.5 from the skull. The infusion was done over eight minutes. Ten minutes after the end of the microinfusion, the needle was removed and an optical fiber (Doric Lenses, diameter 4.8; NA 0.66 Cat # MFC 400/430) within a zirconia ferrule was implanted into the PFC with its tip targeting 0.1 mm above the injection coordinates. Ferrules were secured to the skull using dental cement with the help of two skull-penetrating screws. Mice were left undisturbed in their home cages for 3-weeks to maximize viral expression[28,29]. Mice were group-housed during recovery and across the experiments.

During the fiber photometry experiment, one male mouse from the saline group and one male mouse from the AMPH group, both exposed during adolescence, were excluded due to insufficient signal. Among the remaining subjects, an additional male mouse from the adolescent saline group did not exhibit a distinct saline peak and was therefore omitted from this portion of the analysis. For female mice, data could not be obtained from two individuals treated with AMPH and two treated with saline in adolescence, owing to either lack of signal or loss of the headcap containing the

optical fiber. Regarding animals subjected to CRISPRa, two mice from the LacZ group, one which received saline and another one which received AMPH during adolescence, did not show fluorescent signal. One individual from the *Dcc* overexpression group, which received AMPH during adolescence, was excluded as an outlier from the analysis.

### CRISPRa (clustered regularly interspaced short palindromic repeats activation)
To increase DCC receptor expression in adolescence, we employed CRISPRa with 4 compatible single guide RNAs (sgRNAs) as in ref. 11. Robust neuronal expression was ensured using lentiviral constructs optimized for this purpose (lenti SYN-FLAG-dCas9-VPR, RRID:Addgene_114196; lenti U6-sgRNA/EF1a-mCherry, RRID:Addgene_114199)[30]. A non-targeting sgRNA against the bacterial *LacZ* gene served as a control. For detailed information about construction and validation see ref. 11.

Early adolescent wild-type C57BL/6 (PND21 ± 1) male mice were anesthetized with Isoflorane (Fresenius Kabi, Canada) and then placed in a stereotaxic frame to bilaterally microinfused a total of 1.0 μl of lentiviral mix containing 0.33 μl of four *Dcc* sgRNAs or 0.33 μl of *LacZ* sgRNA and 0.66 μl of the dCas9-VPR virus in sterile PBS[11,30], using the Stoelting Quintessential Stereotaxic Injector attached to a Hamilton 5 μL microsyringe 75RN. The lentiviral mixture was microinjected into the ventral tegmental area (VTA) using the following coordinates: AP: −2.56 mm relative to Bregma, ML: 0.75 mm DV - 4.5 from the skull at 10-degree angle. The infusion was done over 16 min. 10 min after the end of the infusion the needle was removed. Animals were left to recover for 24 h before starting the place preference conditioning.

### Fiber photometry
On PND 101 ± 15 fiber photometry signals were collected using a fiber photometry rig with optical components from Doric. The LED beams were reflected and coupled to a fluorescence minicube (FMC4, Doric Lenses). A 0.5 m-long optical fiber (400 mm, Doric Lenses) was used to transmit light between the fluorescence minicube and the implanted fiber. Lenses were controlled by a real-time processor from Tucker Davis Technologies (TDT; RZ10). TDT Synapse software was used for data acquisition. 465 nm and 405 nm LEDs were modulated at 211 Hz or 230 Hz and at 330 Hz, respectively. LED currents were adjusted to return a voltage between 150 and 200 mV for each signal, were offset by 5 mA, and were demodulated using a 4 Hz low-pass frequency filter. During the experiment, mice were placed in an open field (11–1/2″ Long x 7-1/2″ Wide x 5″ Deep) and attached to the optical fiber. First, they were allowed to habituate to the environment for 1 h. Following a 10-min baseline recording, mice received an intraperitoneal (i.p.) injection of saline, and recording continued for 20 min. Mice were then administered an i.p. injection of methylphenidate (10 mg/kg), and recordings were continued for an additional 90 min.

### Western-blot
For Western blot analysis, we prepared protein lysates of the prefrontal cortex of adult male mice (PND 101 ± 15) exposed to AMPH or saline in early adolescence (PND 21–32) using the CPP paradigm. Mice were left undisturbed in their home cages and at PND101 ± 15 their brains were extracted and flash frozen. Bilateral tissue punches of the PFC were collected as before[11]. We used a RIPA lysis buffer (Milipore-Sigma) containing protease and phosphatase inhibitors (Roche) to prevent protein degradation and dephosphorylation. Protein concentrations were determined using a BCA protein assay kit (Thermo Fisher Scientific), and we normalized samples to ensure equal protein loading. Protein lysates (20 μg) were denatured by heating at 95 °C for 5 min in a Laemmli loading buffer (Bio-Rad) containing β-mercaptoethanol (Sigma) We then separated the proteins by SDS-PAGE (Sodium dodecyl sulfate–polyacrylamide gel electrophoresis) on a 4–15% Mini-PROTEAN TGX Stain-Free Gels (Bio-Rad) at 180 V for 30 min. Following electrophoresis, we transferred the separated proteins to a PVDF (polyvinylidene difluoride) membrane (Bio-Rad) using

a Trans-Blot Turbo transfer system (Bio-Rad, Hercules, CA, USA) at 2.5 A and 25 V for 7 min.

Immunoblotting and Detection: After protein transfer, we blocked the PVDF membrane in a 5% BSA (Millipore-Sigma) in TBST (Tris-buffered saline with 0.1% Tween 20) for 1 h at room temperature to prevent non-specific antibody binding. The membrane was then incubated overnight at 4 °C with polyclonal rabbit anti-VMAT2 1:2500; kindly provided by Dr. G.W. Miller[31], which was diluted in the blocking solution, and with monoclonal rabbit anti-GAPDH (1:10,000, Cell signaling, 3683S).

Following primary antibody incubation, we washed the membrane 3 times for 10 min each in TBST. We then incubated the membrane with Goat Anti-Rabbit IgG Antibody (H + L), Biotinylated (1:5000; vector laboratories, BA-1000-1.5) and streptavidin-HRP (1:5000; Thermofisher, 434323) each for 1 h at room temperature. After three more washes in TBST, we detected the protein bands using a Clarity chemiluminescence (ECL) Western blotting substrate (Biorad) and visualized them using a ChemiDoc XRS+ imaging system (Bio-Rad). We used densitometry to quantify the band intensities using Imagelab (Biorad), and we normalized the protein expression levels to GAPDH, to ensure accurate comparisons between samples. One of the samples obtained from the AMPH treated group was removed from the analysis since it was an outlier.

## Neuroanatomical analysis

**Perfusion**. In adulthood and at the end of the experiment (PND101 ± 15), mice were anesthetized with a cocktail of ketamine 100 mg/kg, xylazine 10 mg/kg, acepromazine 3 mg/kg and perfused with cold PBS followed with 4% paraformaldehyde. The brains were sliced into 35µm coronal sections using a Leica vibratome.

**Immunolabeling**. As previously done, every second coronal brain section was processed for immunofluorescent labeling (1:2 series)[32]. Tissue sections were blocked (2% bovine serum albumin, 0.2% Tween-20 in PBS) for 1 h and subsequently incubated for 48 h at 4 °C with a monoclonal rat anti-dopamine transporter (DAT) antibody (1:500, MAB369, Millipore Bioscience Research Reagents)[33] and a polyclonal rabbit anti-tyrosine hydroxylase (TH) antibody (1:500, AB152, Millipore Bioscience Research Reagents)[34] diluted in blocking solution. Sections were then rinsed with 0.2% PBS-T and then incubated for 2 h at room temperature with Goat anti-Rat Alexa Fluor 488 and Donkey anti-Rabbit Alexa Fluor 594-conjugated secondary antibodies (1:500; Invitrogen, A-11006 and A-21207). Sections were coverslipped onto gelatin-coated slides using a SlowFade Gold Antifade mounting medium (Invitrogen).

## Stereology

The cingulate 1 (Cg1), prelimbic (PrL), and infralimbic (IL) subregions of the pregenual PFC were delineated according to plates spanning 14–18 of the mouse brain atlas (Paxinos and Franklin, 2019) and contours were traced at ×5 magnification using a Leica DM400B microscope along the dense TH-positive innervation of PFC layers V-VI[35]. We assessed the density of DAT+ varicosities in the Cg1, PrL and IL subregions using a stereological fractionator sampling design[36,j] Stereoinvestigator software (MBF, St. Albans VT) as previously[13,35]. Varicosities were counted at 100X magnification.

Two male subjects who received AMPH were excluded from the stereological analysis because their coronal brain sections were damaged during processing and handling for immunofluorescence, making them unsuitable for stereological analysis.

## Verification of viral transduction CRISPRa

Every second brain coronal section was processed for visualization of tyrosine hydroxylase (TH) + neurons and mCherry—the reporter tag added to the sgRNA. Sections were rinsed 3 times for 10 min with 1x PBS and blocked in 2% bovine serum albumin (in 1x PBS and Tween-20) for 1 h at room temperature. Sections were then incubated for 48 h at 4 °C in the primary antibodies: mouse anti-TH (Millipore Sigma, cat. no. MAB318) and rabbit anti-red

fluorescent protein (RFP) for mCherry (Rockland, cat. no. 600-401-379). Sections were rinsed 3 times for 10 min with 1x PBS and incubated in the secondary goat anti-mouse Alexa Fluor 488 antibody (Invitrogen, cat. no. A-11001) and the secondary donkey anti-rabbit Alexa Fluor 594 antibody (Invitrogen, cat. no. A-21207), for 1 h at room temperature. Sections were rinsed 3 times in 1x PBS and mounted with VECTASHIELD Hardset antifade mounting medium with DAPI (Vector Laboratories, cat. no. H-1500-10). Representative images were taken using an epifluorescent microscope (Leica DM400X3).

To label neurons expressing the GRAB_DA senor, tissue sections were rinsed and then immunostained with chicken anti-GFP antibody (1:500, Abcam, Cat#ab13970), followed by the Alexa-488-conjugated goat-anti-chicken (1:200, AAT-Bio, Cat#16687) secondary antibody.

## Data analysis

*CPP data* were analyzed using a three-way repeated measures ANOVA with conditioning test Day (PND32; PND80) as the repeated measure and drug treatment (AMPH; saline) and sex as between subjects factors. Due to the difference in variance in the CPP data obtained in the CRISPRa experiment, these data were analyzed using Generalized Estimating Equations (GEE) with two-independent (sgRNA and Treatment) and one repeated (conditioning tests) factors.

*Density of DAT+ varicosities data* were analyzed using a three-way repeated measures ANOVA, with PFC subregion as the repeated measure and drug and sex as between subjects factors.

*Dopamine dynamics data* were processed using a MATLAB script (https://doi.org/10.6084/m9.figshare.30267643.v1). Noise-related changes in $GRAB_{DA}$ fluorescence across the whole experimental session were removed by scaling the isosbestic control signal (405 nm) and regressing it onto the dopamine-sensitive signal (465 nm). This regression generated a predicted model of the noise that was based on the isosbestic control[37]. Dopamine-independent waveforms on the 405 nm model were then subtracted from the raw $Grab_{DA2h}$ signal. The resulting signal was converted to ΔF/F0 by dividing it to the fitted isosbestic control.

$$\Delta F/F0 = Grab_{DA2h} \text{signal} - \text{fitted isosbestic control/fitted isosbestic control}$$

A linear regression to correct for photobleaching was fitted to data obtained before vehicle injection and applied across the series. For comparison of data across mice, a robust z-score[38] (z ΔF/F0) was computed using the median and the median absolute deviation (MAD) from the 10 min preceding the saline injection.

$$Z - \text{score} = \Delta F/F0 \text{event}(i) - \text{Median Baseline/MAD of baseline}$$

We calculated the area under the curve and used a two-way ANOVA to quantify the effect of treatment (saline or AMPH) and sex.

The baseline transient frequency and amplitude were processed using a MATLAB script (https://doi.org/10.6084/m9.figshare.30267643.v1) and the peak finder procedure written by Nathanael Yoder (2023). This function searches noisy signals for derivative crossings (local maxima/minimal) that are at least a specified amount above or below the last derivative crossing. Peak amplitudes were measured from the last local minimal found by the function.

The dopamine signal observed after the i.p. saline injection was measured using exponential equations that were fitted to the ascending and descending parts of the curve using MATLAB. The ascending part of the curve was fitted to the following equation:

$$Y = Y0^* \exp(k^*X).$$

The descending part was fitted to the one phase decay equation:

$$Y = (Y0 - Yplateau)^* \exp(-k^*X) + Y.$$

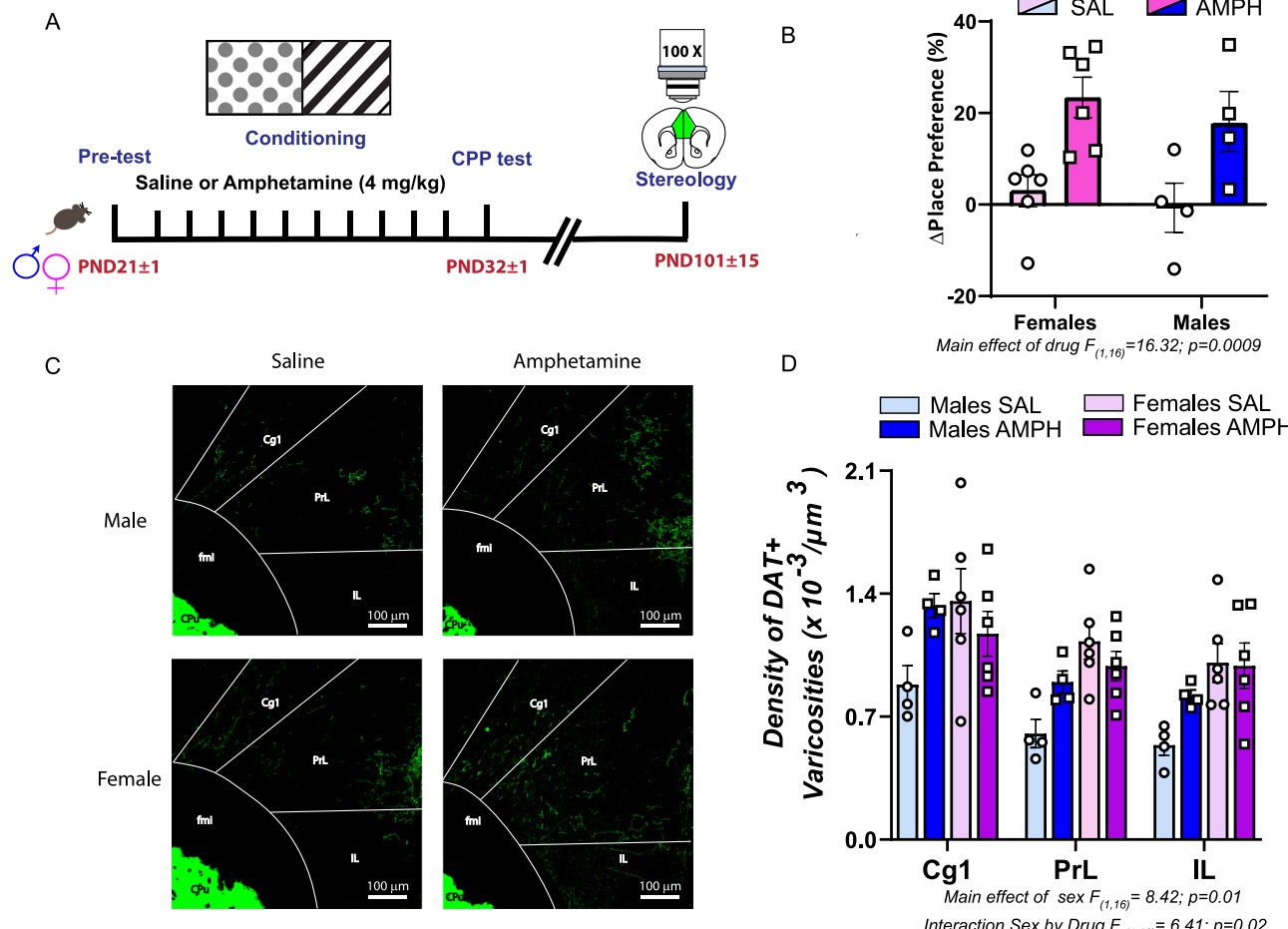

**Fig. 1 | Rewarding doses of amphetamine (AMPH) in adolescence increase the density of dopamine transporter (DAT)+ axons in the adult PFC of male, but not female mice. A** Experimental timeline. **B** Male and female mice exposed to AMPH (4.0 mg/kg) in early adolescence develop place preference for the side of the box paired with the drug. **C** Photoimages of representative coronal sections of DAT+ axons in the cingulate (Cg1), prelimbic (PrL) and Infralimbic (IL) subregions of the PFC of adult male and female mice exposed to AMPH or to saline in adolescence. In males, AMPH in adolescence increases the number of DAT+ axons compared to saline controls (*Top*). Adult females display similar levels of DAT+ axons regardless of adolescent treatment (*Bottom*). In saline control groups, females exhibit higher number of DAT+ varicosities than males across the three PFC subregions. **D** Stereological analysis revealed increased density of DAT+ varicosities in adult males exposed to AMPH in adolescence compared to their saline counterparts. In females, no differences between AMPH and saline groups were found. In saline pretreated groups, adult females have higher density of DAT+ varicosities, compared to males. Sample size: Female Saline $n = 6$; Female AMPH $n = 6$; Male Saline $n = 4$; Male AMPH $n = 4$. Total $N = 20$. Descriptive statistics and statistical tests performed are described in detail in Supplementary Data 1. The underlying source data for the graphs is presented in Supplementary Data 2.

In these equations Y0 is the Y value when X (time) is zero; Yplateau is the value of Y at infinite times, k is the rate constant.

The parameters obtained were then analyzed using a two-way ANOVA with treatment in adolescence (saline, AMPH) and sex as independent factors.

In all the statistical analyses, Holm-Šídák's multiple comparisons tests were used when significant interactions were detected. All descriptive statistics and statistical tests performed are described in the Supplementary Data 1. The underlying source data for the graphs is presented in the Supplementary Data 2. When significant, the results of the statistical analysis are embedded in the figures. Statistical significance was defined as $p \leq 0.05$. Normality was assessed using Shapiro-Wilk tests. Experimenters were blinded to group allocation during neuroanatomical analysis. All statistical analyses were carried out using Prism software (GraphPad), except for the GEE analysis, which was done in SPSS.

## Reporting summary

Further information on research design is available in the Nature Portfolio Reporting Summary linked to this article.

## Results

### AMPH in adolescence increases the density of DAT+ axons in the adult PFC of male, but not female mice

Mesolimbic and mesocortical dopamine axons have distinct anatomical, molecular and functional properties[19,20]. In male mice, early adolescent exposure to AMPH reroutes Nac dopamine axons to the PFC[11]. To test whether ectopic mesolimbic dopamine axons retain the molecular phenotype of their intended target or adopt those of the PFC, we quantified the density of DAT+ varicosities in the adult PFC of mice exposed to AMPH or saline in early adolescence (Fig. 1A). We performed this experiment in male and female mice to assess for potential sex differences.

AMPH in early adolescence induces robust and comparable levels of CPP in both males and females, 24 h after the last conditioning session, in line with our previous report[11] (Fig. 1B). In adulthood, AMPH-pretreated males but not females show a robust increase (~50%) in the density of DAT+ varicosities across the Cg1, PrL and IL PFC subregions, compared to their saline counterparts (Fig. 1C, D). This indicates that misrouted mesolimbic dopamine axons maintain molecular signatures of their intended target.

Notably, in the saline control groups, females show higher PFC DAT+ varicosities density (55%) compared to males. To our knowledge, this is the first demonstration of increased DAT expression in the PFC of males compared to females. Our findings suggest sex differences in dopamine reuptake in this region, which may translate into differences in PFC function and cognitive processing.

### AMPH in adolescence does not increase vesicular monoamine transporter 2 (VMAT2) expression in the adult PFC of male mice

In addition to the reuptake of extracellular dopamine by DAT, dopamine is accumulated and compartmentalized in cytoplasmic vesicles by VMAT2, thereby regulating dopamine concentrations within neuronal compartments[39,40]. To verify if AMPH in adolescence alters the expression of VMAT2 in the adult PFC of male mice, we used western immunoblot of PFC tissue of adult males mice exposed to AMPH or saline in adolescence. The VMAT2 antibody detected two bands at approximately 70 and 56 kDa, in agreement with previously reported molecular weights for this protein following post-translational modifications in mice[31,41]. The levels of adult VMAT2 isoforms in the PFC do not differ between AMPH and saline treated groups (Supplementary Fig. 1).

### AMPH-induced CPP incubates during the adolescent period

We next examined if AMPH in early adolescence leads to long-term CPP and if this effect is sex-specific (Fig. 2A). As expected, AMPH treatment induced preference for the drug-paired compartment 24 h after the last conditioning session. This preference persisted and became more prominent in adulthood (Fig. 2B), revealing an incubation-type effect and suggesting that the rewarding effects of AMPH are latent and strengthen with age[42].

### AMPH in adolescence leads to exaggerated dopamine responsiveness in the adult PFC of males

To explore the functional consequences of altered PFC dopamine innervation in adult males exposed to AMPH in early adolescence, we investigated dopamine transient kinetics and dynamics using fiber photometry. Twenty-four hours after the adult CPP test, AMPH- and saline-pretreated male and female mice underwent stereotaxic unilateral microinfusion of AAV- hSyn-rDA2h (Grab$_{DA2h}$) into the IL part of the PFC, followed by intracranial implantation of an optical fiber. Three weeks later, we quantified dopamine signals at baseline, following an i.p. injection of saline and following an acute i.p. administration of methylphenidate at a 10 mg/kg dose (Fig. 2A).

**Baseline**. Using the results obtained from the peak finder procedure[43] we compared the number and amplitude of dopamine transients during the ten-minute baseline, right after the 1 h habituation period (Fig. 2C, D, red square). We found that adult male mice treated with AMPH in early adolescence have a reduced number of spontaneous dopamine transients compared to saline controls, most likely due to increased dopamine reuptake. This effect of AMPH is not observed in females. There are no sex differences in the baseline number of dopamine transients (Fig. 2E). Interestingly, in males, AMPH administered in adolescence *increases* baseline dopamine transient amplitude in adulthood (Fig. 2F). This may be the result from a combined effect of reduced transient frequency and increased amount of readily releasable dopamine due to higher density of DAT+ varicosities in the PFC. In females, there are no differences between AMPH and saline groups. Surprisingly, there are no differences in baseline dopamine transient amplitude between males and females despite the sex differences in DAT expression (Fig. 2F).

**Saline**. If in AMPH-pretreated males, dopamine accumulation at terminals is enhanced during baseline conditions due to an increase in dopamine recycling, this group should show accelerated dopamine release and clearance in response to an acute i.p. injection of saline (Fig. 3A, B). We assessed this possibility by fitting a double logarithmic

function to the ascending and descending parts of the fluorescent peak. Indeed, in adult males, an acute saline injection induces faster dopamine release in the AMPH- versus the saline-pretreated group. Accelerated dopamine release is indicated by both the steeper slope (k ascending) (Fig. 3C) and the shorter time it takes for the fluorescence signal to double (doubling time) (Fig. 3D). In females, the timing of dopamine release does not differ between AMPH- and saline-pretreated groups.

In both male and female mice, we were unable to capture any difference in the fluorescent signaling used to infer reuptake (Fig. 3E, F) since both groups show similar slope (k descending) and half-life parameters. This result is surprising because we expected a faster dopamine clearance in AMPH-treated males due to their increased DAT expression. The high-affinity DA sensor used, which provides high affinity but has a slow signal decay time (8.3 s)[44,45], could be masking the detection of rapid clearance events.

**Methylphenidate**. Previous work has shown that DAT overexpression increases methylphenidate-induced increase in extracellular dopamine, at least in the striatum[46]. One key role of DAT is to facilitate the reuptake of dopamine from the extracellular space back into the presynaptic terminal, thereby enabling its reuse[47]. However, dopamine terminals in the PFC are relatively inefficient at reuptake due to low DAT density[48]; with the norepinephrine transporter primarily handling this function[49,50]. This inefficiency results in reduced dopamine recycling and reserve pools, with most of the dopamine available for release being newly synthesized[51].

We assessed dopamine signaling in male and female mice after a methylphenidate injection. We anticipated that DAT overexpression in males exposed to AMPH in adolescence would result in exaggerated methylphenidate-induced dopamine signaling due to the increased density of DAT+ varicosities in the PFC. We also anticipated sex-specific methylphenidate-induced dopamine signals, considering the increased density in DAT+ varicosities in females compared to males.

Acute methylphenidate increased dopaminergic signaling across all groups. However, males that received AMPH in adolescence showed significantly higher dopamine release 10 min after the methylphenidate challenge. This signal peaked 30 min after the injection and returned to baseline levels 80 min after (Fig.4A). In contrast, female mice exposed to AMPH or saline in adolescence showed similarly elevated dopamine levels in response to methylphenidate (Fig. 4B). Analysis of the area under the curve confirmed that males pretreated with AMPH, but not their female counterparts, have exaggerated dopamine increase in response to methylphenidate (Fig. 4C). These findings indicate that ectopic dopamine innervation renders the male PFC hypersensitive to methylphenidate-induced dopamine release. Contrary to our prediction, there were no differences in dopamine signaling between male and female groups that received saline exposure in adolescence.

### DCC receptor upregulation in adolescence prevents AMPH-induced place preference in adolescence

We next evaluated whether increasing DCC receptor expression in adolescence, via CRISPRa, prevents changes in PFC dopamine dynamics observed in adult males exposed to AMPH in adolescence. As previously shown[11], we used a combination of four *Dcc* sgRNAs, which increases *Dcc* mRNA up to a four-fold in dopamine cell culture. This construct, when administered into the VTA of adolescent mice, produces a significant increasein DCC protein expression in axons of dopamine neurons in the Nac, compared to *LacZ* sgRNA control infection[11].

Male mice were injected with the sgRNA cocktail and dCas9 viruses at PND21 and then exposed to CPP. Place preference was assessed 24 h and 50 days after the last injection (Fig. 5A). After the last preference test, mice were microinjected with a virus expressing Grab$_{DA2h}$ and an optical fiber aimed at the IL area of the PFC. Fiber photometry assessment of dopamine release was done 3 weeks after.

**Fig. 2 | In males only, AMPH in early adolescence increased PFC dopamine accumulation at baseline in adulthood. A** Experimental timeline. **B** Male and female mice exposed to AMPH (4.0 mg/kg) in early adolescence developed place preference for the side of the box paired with the drug. This effect increased over time, suggesting an incubation-like effect (inset). Sample size: Female Saline $n = 9$; Female AMPH $n = 11$; Male Saline $n = 10$; Male AMPH = 10. Total $N = 40$. **C, D** Average PFC dopamine dynamics during baseline and following an acute i.p. injection of saline and of methylphenidate. The red rectangle highlights the baseline period used in subsequent analyses. **C'- D'.** Representative dopamine traces from adult males and females exposed to saline or AMPH in adolescence. Red circles indicate individual dopamine transients. **E** Males administered AMPH in adolescence show reduced number of dopamine transients during baseline. This reduction is not seen in females. **F** The amplitude of dopamine transients is significantly greater in AMPH-treated males compared to their saline counterparts. Females displayed similar transient amplitude regardless of adolescent treatment. All bar graphs are presented as mean values ± SEM. Sample size: Female Saline $n = 6$; Female AMPH $n = 6$; Male Saline $n = 8$; Male AMPH = 8. Total N = 28. Descriptive statistics and statistical tests performed are described in detail in Supplementary Data 1. The underlying source data for the graphs is presented in Supplementary Data 2. *$p < 0.05$; **$p < 0.01$. ADO adolescence.

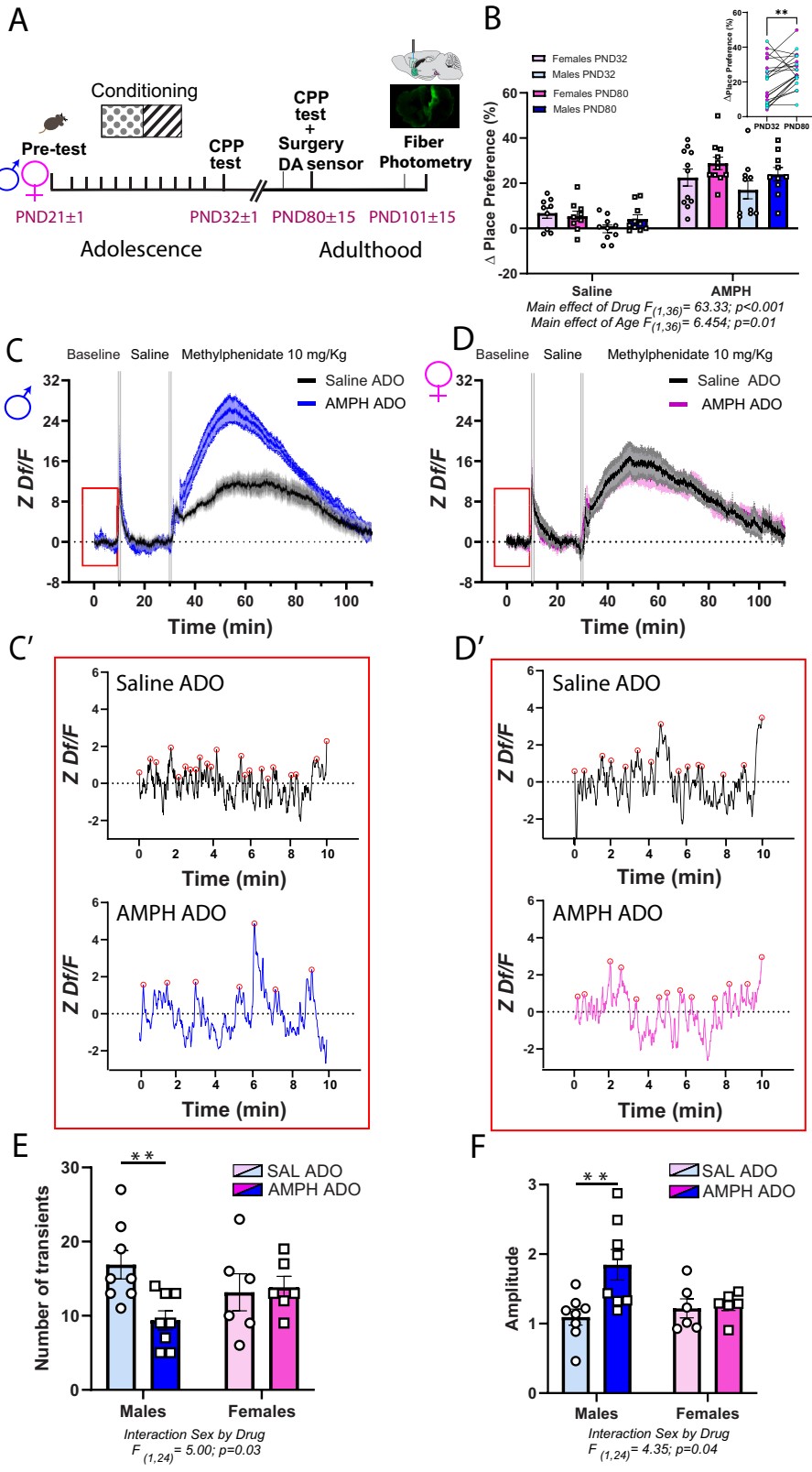

Adult male mice exposed to AMPH in adolescence and infected with *LacZ* sgRNA showed robust CPP 24 h after the last conditioning session. The preference for the drug-paired compartment is maintained when mice reached adulthood. Remarkably, DCC receptor overexpression abolished AMPH-induced place preference both 24 h after the last conditioning session and when mice were re-tested in adulthood, although higher preference variability emerged at PND80 (Fig. 5B). These results indicate a role of DCC receptors in AMPH-induced place preference and may be related to changes in drug-induced behavioral plasticity initiated by somatodendritic dopamine release. Indeed, our previous work in adult mice shows that DCC receptor function in VTA dopamine neurons is necessary for the development of AMPH-induced behavioral sensitization[52].

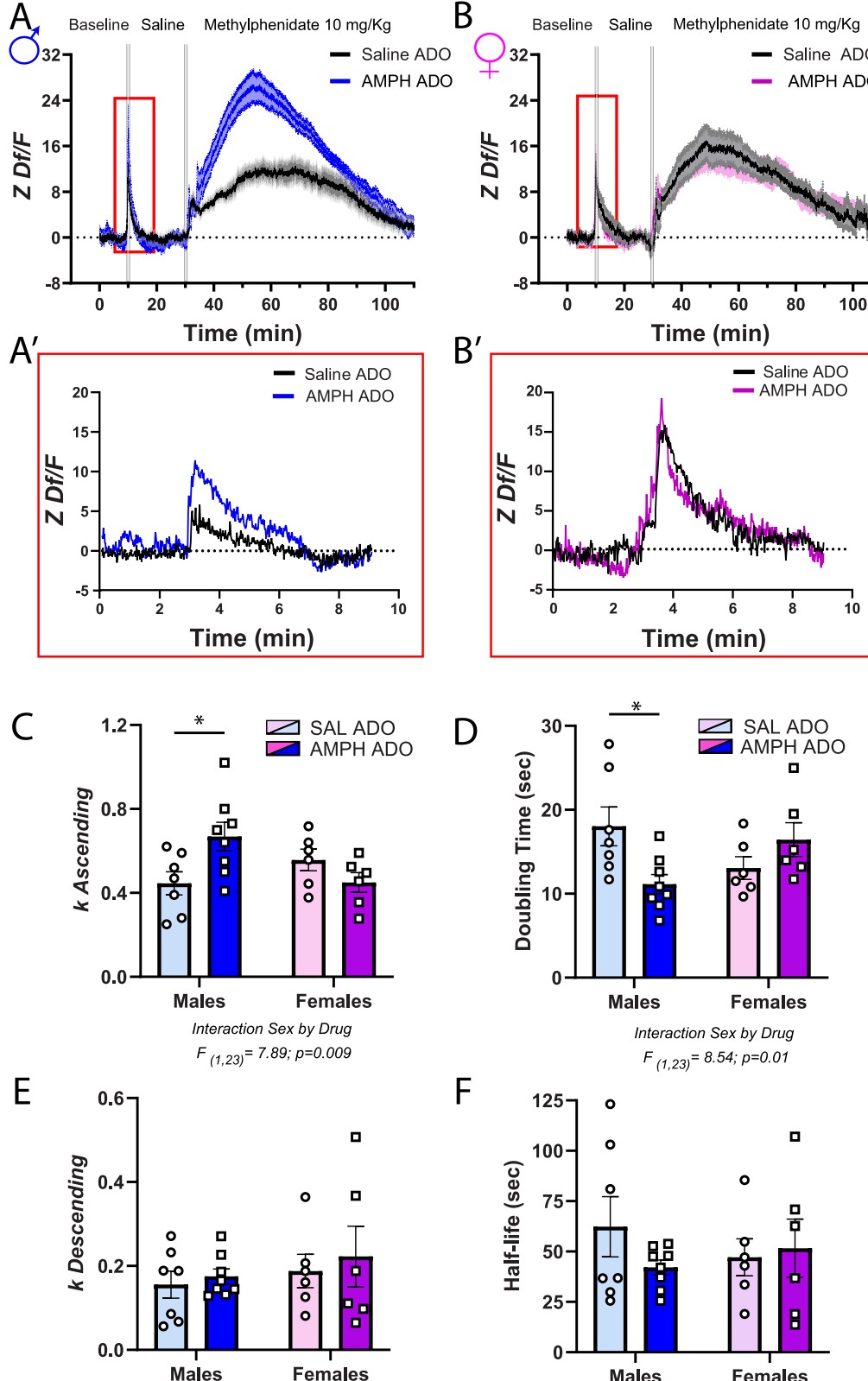

## DCC receptor upregulation in adolescence prevents AMPH-induced changes in adult PFC dopamine release in adulthood

The fiber photometry recordings revealed subtle changes in baseline dopamine activity in the PFC following adolescent AMPH exposure in mice that were transfected with LacZ sgRNA and *Dcc* sgRNA (Fig. 5D-G, D', E'), but these changes did not reach statistical significance.

Unlike in our previous fiber photometry experiment, we could not reliably detect a peak in response to the saline injection. In fact, in the mice that were transfected with the *Dcc* sgRNA, only 3 out of 10 mice (30%) showed a saline-induced peak. This contrasts with the *LacZ* sgRNA group, in which 9 out of 11 animals showed the stress-induced peak (82%). This difference is intriguing, but we do not know the underlying causes.

**Fig. 3 | AMPH in adolescence accelerates PFC dopamine dynamics following an injection of saline in adulthood, but only in males. A**, **B** Average PFC dopamine dynamics during baseline and following an acute i.p. injection of saline and of methylphenidate. The red rectangle highlights dopamine release following the saline challenge. **A'-B'.** Representative dopamine traces from adult males and females exposed to saline or AMPH in adolescence. Peak analysis was performed on each trace after the acute i.p. injection of saline. To each subject, we fitted two exponential curves, one for the ascending part and one for the descending part so as to calculate the slope of each peak ($k$); the time for the signal to double in size (doubling time) and the time the signal decreases in intensity by half (half-life). **C** The $k$ value for the ascending curve of AMPH-pretreated males is significantly higher than that of their saline counterparts, suggesting a faster surge of dopamine release in adult males exposed to AMPH in adolescence. In females, the slopes for the ascending curve are similar between treatment groups. **D** Due to the more rapid increase in dopamine release in AMPH-pretreated males, the resulting doubling time is significantly shorter compared to that observed in males exposed to saline in adolescence. In females, the doubling time is similar regardless of adolescent treatment. **E** The slopes observed in the descending curves of both male and female groups demonstrate a consistent pattern, indicating a comparable rate of decline in dopamine concentration across all groups. **F** For the half-lives, there are no differences across groups. All bar graphs are presented as mean values ± SEM. Sample size: Female Saline $n = 6$; Female AMPH $n = 6$; Male Saline $n = 7$; Male AMPH = 8. Total $N = 27$. Descriptive statistics and statistical tests performed are described in detail in Supplementary Data 1. The underlying source data for the graphs is presented in Supplementary Data 2. *$p < 0.05$. ADO adolescence.

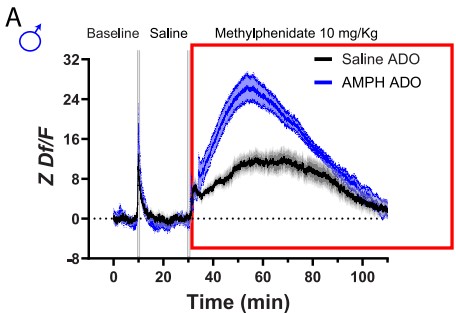
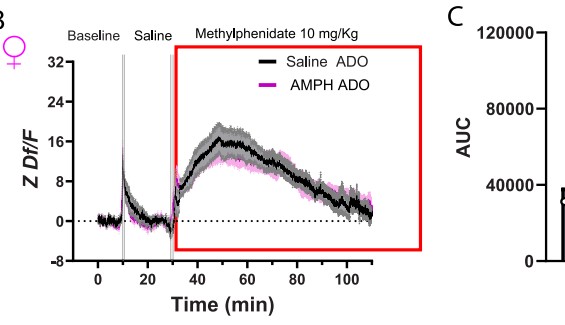

**Fig. 4 | AMPH in adolescence leads to exaggerated methylphenidate-induced increase in PFC dopamine release in adult males only. A** Average PFC dopamine dynamics during baseline and following an acute i.p. injection of saline and of methylphenidate. The red rectangle highlights dopamine release following acute i.p. methylphenidate challenge. Methylphenidate (10 mg/Kg) increases PFC dopamine release in both sexes, but this effect is significantly amplified in males with early adolescent AMPH exposure. **B** In females, methylphenidate also increases PFC dopamine release, but this increase is similar in AMPH and saline groups. **C** These observations are corroborated by quantifying the area under the curve (AUC). All bar graphs are presented as mean values ± SEM. Sample size: Female Saline $n = 6$; Female AMPH $n = 6$; Male Saline $n = 8$; Male AMPH = 8. Total $N = 28$ Descriptive statistics and statistical tests performed are described in detail in Supplementary Data 1. The underlying source data for the graphs is presented in Supplementary Data 2.**$p < 0.01$. ADO adolescence.

Upon methylphenidate challenge, mice infected with *LacZ* sgRNA and exposed to AMPH in adolescence showed exaggerated dopamine responsiveness, compared to mice pre-exposed to saline (Fig. 5C). Overexpression of DCC receptors in adolescence completely abolishes this effect, indicating that AMPH-induced downregulation of DCC receptors in adolescence drives dopamine hyper-responsiveness in the PFC in adulthood, most likely due to the presence of ectopic mesolimbic dopamine axons expressing high levels of DAT +. This female-like protective effect, coupled with the absence of AMPH-induced place preference, highlights DCC overexpression as a promising therapeutic target to mitigate short- and long-term consequences of exposure to rewarding doses of AMPH in adolescence. Immunohistochemistry confirmed robust sgRNA expression within the VTA, co-expressed in TH+ neurons (Fig. 5H).

## Discussion

Misuse of stimulants drugs in adolescence – both illicit and prescribed – is a growing concern[53] due to its association with heightened psychiatric risk, including substance use disorders[54–57], increase suicidal ideation[57], and depression[55]. The underlying mechanisms and the causes of observed differences in vulnerability between adolescent boys and girls remain unknown. In male mice, AMPH in early adolescence reduces the expression of the guidance cue receptor DCC in mesolimbic dopamine axons, disrupting their ongoing targeting in the NAc and inducing their ectopic growth to the PFC[11]. Females, however, are protected against these effects. In this study, we assessed the phenotype and function of dopamine axons in the adult PFC of male and female mice exposed to AMPH in adolescence. We found that recreational-like doses of AMPH in adolescence lead to exaggerated density of DAT+ fibers in the PFC of adult males, indicating that misrouted mesolimbic dopamine axons retain molecular and functional properties of their intended target[19–21,58,59]. Our findings also show that adult PFC dopamine signaling in male mice exposed to AMPH in adolescence retain mesolimbic-like characteristics, including larger baseline dopamine transients and exaggerated release in response to acute stress and stimulants. Restoring DCC receptor levels in adolescent males via CRISPRa-targeted gene therapy prevents disruptions in dopamine development. In contrast, the same recreational-like doses of AMPH in females do not alter PFC dopamine phenotype in adulthood, highlighting the male-specific vulnerability to dopamine development disruption in early adolescence by drugs of abuse.

The increase in DAT expression in PFC dopamine terminals likely mediates the alterations in dopamine signaling observed at baseline and in response to the acute challenge of saline and methylphenidate in male mice. The DAT protein has been shown to regulate the timing, strength, and overall function of dopamine signaling in the striatum and in cultured neurons[60], including the reuptake and repackaging of dopamine after being released[61]. Increased DAT levels in the striatum have been shown to enhance dopamine recycling and reserve pools, reducing the use of newly synthesized dopamine for release[51]. The larger PFC dopamine transients observed at baseline in AMPH-pretreated male mice compared to saline controls is likely to reflect heightened dopamine reuptake, reserve pools, and recycling due to greater PFC DAT+ expression. The fact that the dopamine transients are more sporadic may result from increased dopamine autoreceptor activation, which would decrease neuronal excitability and reduce the probability of dopamine release under baseline conditions[62].

There is some evidence suggesting that DA axons in the NAc and in the PFC also have different levels of the vesicular transporter VMAT2[22,63]. We assessed if the impact of AMPH in adolescence on DAT expression in the adult PFC in male mice would also extend to changes in VMAT2 levels. We did not observe changes in PFC VMAT2 levels in AMPH-treated mice, possibly due to the limitation of the Western blot analysis to detect subtle changes in protein levels within axon terminals. Another potential factor is the expression of VMAT2 in astrocytes within the PFC[64], which may mask

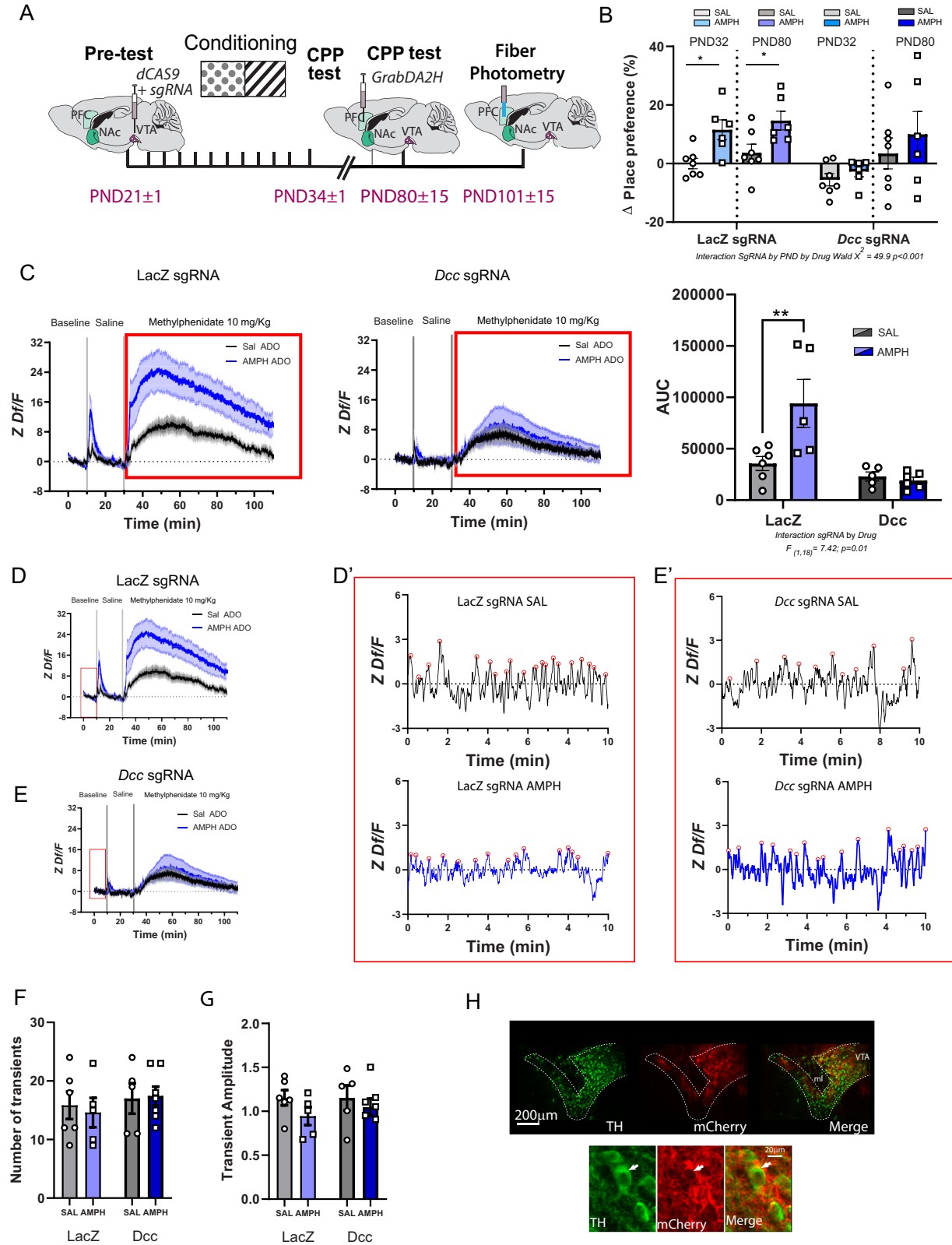

any potential changes occurring in dopamine terminals. Future experiments should include stereological quantification of VMAT2+ axons to shed light into this issue.

Dopamine function in the PFC is highly sensitive to stress[65,66]; we were able to measure group differences in stress-induced changes in dopamine transients via an acute i.p. injection of saline. The steeper and faster

dopamine release we observed in response to saline in AMPH-pretreated male mice, compared to saline counterparts, is consistent with these mice having greater PFC DAT expression and, therefore, increased dopamine reserve pools. The fact that we did not capture differences between groups in stress-induced dopamine clearance is likely due to the slow signal decay of the Grab$_{DA2h}$ sensor, which cannot detect differences in fast dopamine

**Fig. 5 | Increasing *Dcc* transcription in the VTA in adolescence with CRISPRa, prevents PFC dopamine dysfunction in adult males exposed to AMPH in adolescence. A** Experimental timeline. **B** AMPH induces conditioned place preference in the *LacZ* sgRNA group in adolescence and this preference persists into adulthood. Increased VTA *Dcc* transcription in adolescence impairs AMPH-induced place preference in adolescence and in adulthood. Sample size: LacZ sgRNA Saline *n* = 7; AMPH = 6. *Dcc* sgRNA Saline *n* = 7; AMPH = 6 Total *N* = 26. **C** AMPH in adolescence increases adult PFC dopamine release in response to the methylphenidate (10 mg/kg) challenge in adulthood in the *LacZ* sgRNA group compared to their control counterparts. This effect is completely prevented in mice with *Dcc* upregulation in adolescence. The area under the curve analysis corroborates this finding. **D, E** Average PFC dopamine dynamics during baseline and following an acute i.p. injection of saline and of methylphenidate in males with CRISPRa-induced *LacZ* (**C**) or *Dcc* (**D**) sgRNA transcription. The red rectangle highlights the baseline period

used in subsequent analyses. **D'- E'.** Representative dopamine traces from *LacZ* and *Dcc* sgRNA groups exposed to AMPH or saline in adolescence. Red circles indicate individual dopamine transients. **F** There is no difference in the number of transients during the baseline period between *LacZ* and *Dcc* sgRNA groups regardless of treatment in adolescence. **G** There is no difference in the transient amplitude between *LacZ* and *Dcc* sgRNA-groups regardless of treatment in adolescence. **H** Photoimages of coronal VTA sections showing low and high magnification of dopamine neurons (TH+ green) co-expressing the sgRNA viruses (mCherry). Arrowhead denotes colocalization. All bar graphs are presented as mean values ± SEM. Sample size: LacZ sgRNA Saline *n* = 6 ; AMPH = 5. *Dcc* sgRNA Saline *n* = 5 ; AMPH = 6 Total *N* = 22. Descriptive statistics and statistical tests performed are described in detail in Supplementary Data 1. The underlying source data for the graphs is presented in Supplementary Data 2.**$p < 0.01$. ADO adolescence.

clearance events[44,45]. While a faster technique like Fast-Scan Cyclic Voltammetry could potentially capture these dynamics, its use in the PFC is complicated by the difficulty of distinguishing dopamine signals from those of co-released norepinephrine[67]. However, the stereological data showing increased DAT expression in the PFC of adult male mice exposed to AMPH in adolescence, compared to their saline counterparts, is in line with the accelerated dopamine release observed in this group, manifested as a steeper slope (k ascending) and a shorter time needed for the fluorescence signal to double (doubling time).

The exaggerated increase in PFC dopamine signal following the methylphenidate challenge in adult males treated with AMPH in adolescence is also consistent with the presence of DAT+ ectopic mesolimbic dopamine axons in this region and indicates increased potency of methylphenidate. Heightened dopamine function has been observed in the PFC of adult rats overexpressing DAT(DAT-tg)[68] and in the NAc of adult transgenic mice overexpressing DAT(DAT-tg) in response to methylphenidate[69]. Elevated DAT expression in the NAc of rats with a history of methylphenidate self-administration also leads to enhanced drug-induced dopamine release[46,69]. Methylphenidate not only binds to DAT but also interacts with VMAT2, a protein that packages dopamine into vesicles for release[70,71]. The increased density of DAT in the PFC of males exposed to AMPH in adolescence likely amplifies this effect by clearing dopamine from the synapse and generating a concentration gradient that favors further dopamine release from the nerve terminal. These results suggest that the mesolimbic-like PFC phenotype in these mice is a consequence of the misrouting of NAc dopamine axons to this region. The changes in PFC dopamine function due to early adolescent AMPH administration can be seen as a risk factor leading to drug abuse, not only because it may be linked to impaired behavioral inhibition[11,72], but also because of the enhanced potency of methylphenidate and potentially of other drugs of abuse that increase dopamine release[69].

In contrast to males, female mice exposed to AMPH in adolescence do not exhibit elevated DAT levels in the PFC or altered dopamine at baseline, or in response to saline or methylphenidate. This sexual dimorphism suggests that aberrant PFC dopamine dynamics in males result from ectopic mesolimbic dopamine innervation. Unlike males, females are resilient to the disruptive effects of early adolescent AMPH on microRNA-related processes[11] and do not show *Dcc* mRNA reduction. By mid-adolescence, females show microRNA epigenetic changes and *Dcc* mRNA reduction, but also exhibit compensatory increases in the expression of the DCC receptor ligand, Netrin-1, maintaining normative mesolimbic dopamine axon targeting[11].

Adult female mice exposed to saline in adolescence show significantly higher baseline DAT expression in the PFC compared to males–a sex difference that has been reported in the ventral and dorsal striatum of mice[73] and humans[74,75]. Why adult females have higher PFC DAT levels than males at baseline, is a question that needs to be addressed, but likely involves sexual dimorphisms in the organization of mesocorticolimbic dopamine circuitry[76]. In contrast with adult male rats, females have a larger proportion of VTA dopamine neurons projecting to the PFC[77], as well as more dopamine neurons and a greater VTA volume[78]. Interestingly, adult males and

ovariectomized adult female rats have lower levels of DAT expression in the VTA and striatum, compared to intact females[79–82]. Despite the overall increase in PFC DAT in female mice, we did not observe sex differences in dopamine dynamics. One reason could be that there are sex differences in mechanisms regulating dopamine release in the PFC. For example, while blockade of NMDA receptors in adult male rats increases extracellular dopamine levels in the PFC[83], this manipulation reduces dopamine release in females[84]. Another possible mechanism is the regulation of DAT function in the PFC by D2 autoreceptors– a process that has been shown to be sex- and circuit-dependent in the striatum[85].

Both male and female mice exhibit robust and long-lasting AMPH conditioned place preference, indicating similar rewarding and positive drug associations[86]. Additionally, both sexes showed an incubation-like effect, suggesting that the neuroanatomical changes underlying this process are similarly affected in males and females. Resilience in females to AMPH-induced disruption of dopamine development does not imply vulnerability in other domains contributing to addiction. Indeed, while the initial experience with the drug may be similar between sexes, the long-term consequences and vulnerability to addiction may diverge. Preclinical models of addiction show that, in general, female rats acquire self-administration of drugs and escalate their drug intake with extended access more rapidly than males, show more motivational withdrawal, and show greater reinstatement[87]. Females also show more motivation to work for AMPH[88]. In humans, a similar pattern is observed: while men are more likely than women to use almost all types of illicit drugs[89], women tend to escalate more rapidly than men, and once they develop substance use disorder, it is more difficult for females to cease drug use.

Increasing *Dcc* gene expression in the VTA of male mice during adolescence, using CRISPRa, prevented the exaggerated dopamine response to methylphenidate in adulthood. This supports the idea that the mesolimbic-like dopamine phenotype in the PFC is caused by AMPH-induced rerouting of NAc dopamine axons to the PFC and that changes in dopamine axon branching are not involved[11,13,18]. Restoring DCC levels also prevented the expression of conditional place preference, suggesting changes in reward sensitivity to AMPH. This effect may be mediated by DCC receptor modulation of behavioral plasticity resulting from somato-dendritic dopamine release. This idea is supported by our previous work demonstrating that DCC receptor function in the VTA is required for the development of AMPH-induced behavioral sensitization in adult mice[52]. DCC has also been shown to play a critical function in adult brain plasticity, via the orchestration of neuronal circuitry reorganization[90].

In this study, we demonstrated that AMPH exposure during adolescence in mice leads to male-specific alterations in dopamine signaling in the adult PFC. This effect is likely due to the mistargeting of NAc dopamine axons to the PFC. These ectopic axons seem to adopt molecular and functional properties of mesolimbic-projecting dopamine neurons, including increased DAT expression and exaggerated responses to methylphenidate.] Dopamine neurotransmission in the PFC plays a crucial role in higher-order cognition[91–95]. The alterations in PFC dopamine signaling induced by AMPH in adolescence may underlie the risk for cognitive

deficits observed in disorders of poor impulse control, including drug addiction[64,96–99]. Our results indicate that DCC receptor upregulation, via pharmacological or non-invasive innervation, may be a promising target for therapeutic intervention[14,100,101].

## Data availability

The data supporting the finding as the MATLAB code are openly available in figshare https://doi.org/10.6084/m9.figshare.30267817.v1. https://doi.org/10.6084/m9.figshare.30267643.v1.

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

## Acknowledgements

This work was supported by the National Institute on Drug Abuse at the National Institutes of Health (R01DA037911 to C.F.), the Canadian Institutes of Health Research (PJT 190045; FRN:17030, 170130 to C.F.) and the Natural Sciences and Engineering Research Council of Canada (RGPIN-2020-04703 to C.F.; CGS-M to T.C.). The present study used the services of the Molecular and Cellular Microscopy Platform in the DHRC. Melina Jaramillo Garcia and Bita Khadivjam helped set up the imaging experiments. We thank Dr. Jorge Quillfeldt for his insightful comments on the manuscript. We thank Dr. Louis Eric Trudeau for providing the VMAT-2 antibody. The timeline diagrams included in Figs. 1, 2, 5 and Supplementary 1 were created using Illustrator software. The cartoons of the mouse brains shown in Figs. 2, 5 were adapted from the ones made by Dr. Lauren Reynolds and presented in our previous publication[11].

## Author contributions

G.H. and C.F. designed the study. G.H., A.S. and S.G. performed behavioral experiments. G.H., J.Z., D.M. and T.C. performed fiber photometry experiments. J.Z., Z.N., T.C., A. Moiz, A. Mahmud, I.H. and G.H. performed neuroanatomy experiments. J.J.D. provided CRISPRa virus material and the sgRNAs. G.H. and C.F. analyzed the data with the contribution and advice of NXT. The writing of the manuscript was done by G.H. and C.F.. All authors discussed the results, edited, and approved the manuscript.

## Competing interests

The authors declare no competing interests.
