## [Transparent Peer Review file · Communications Biology]

Amphetamine in Adolescence Induces a Sex-Specific Mesolimbic Dopamine Phenotype in the Adult Prefrontal Cortex

Corresponding Author: Dr Cecilia Flores

Version 0:

Reviewer comments:

Reviewer #1

(Remarks to the Author)

In this study, the authors investigate whether adolescent amphetamine (AMPH) exposure for 12 days (PND21-32) impacts dopamine (DA) dynamics and the morphological and functional phenotype of DA axons in the medial prefrontal cortex (PFC) of young adult mice. This study builds on the fascinating body of work from this group adolescent demonstrating decreased expression of DCC and increased dopamine transporter (DAT) -expressing neurons in the medial prefrontal cortex (PFC) of male, but not female, mice. In the current study, they also discovered that there is a significantly greater expression of DAT+ axons in control male vs. female mice. After adolescent AMPH, PFC DA axons display abnormal DA transients, faster DA release, and exacerbated responses to the DAT inhibitor, methylphenidate, in young adulthood. Further, CRISPR-mediated upregulation of the axon-guiding Netrin-1 receptor, downregulated in colorectal cancer (DCC), in adolescence prevents these changes. They conclude that ectopic PFC DA neurons in PFC rerouted from terminating in the nucleus accumbens (NAc) in adolescence retain the anatomical and functional phenotypes of NAc axons. The study is novel, appears to be well-performed, and the methods and data are well-described/illustrated, particularly the detailed statistical analysis section, but the discussion and conclusions regarding DAT expression as the primary mechanism underlying these changes are somewhat speculative and lack experimental support, weakening the study's overall impact.

1. DA reuptake/clearance was not significantly altered in adolescent male mice. The data using GRABDA2H which has a slow signal decay time did not detect a change in DA clearance in AMPH-treated male mice. A traditional method to measure clearance should be used to support their conclusion that increased DAT expression underlies the change in DA dynamics.
2. Why wasn't VMAT2 measured to explore whether DA release changes emerge from changes in vesicular release?
3. The authors should investigate whether adolescent methylphenidate has the same effect as AMPH (an experiment not required for this study).
4. Without data from Points 1 and 2, the discussion remains speculative and the conclusions weakened.

Minor Points

1. Introduction-The first paragraph cites studies done in a variety of unidentified species. Please identify the different species in the literature cited.
2. When referring to AMPH doses in rodents throughout the manuscript, please refer to rewarding but not "recreational" doses that are only relevant to humans.
3. Second sentence of second paragraph to improve clarity: Which region are you referring to here "...reduced number of varicosities in this region"? Which region-PFC or NAc?

Reviewer #2

(Remarks to the Author)

The manuscript by Hernandez and colleagues describes an interesting study showing that repeated amphetamine administration, which induces CPP in male and females, leads to an increased expression of prefrontal cortex (PFC) DAT in male, but not female mice. There is also a corresponding increase in dopamine amplitude and transients at baseline, and

increased amplitude following saline and methylphenidate administration in male, but not female, PFC. Finally, amplification of DCC expression via CRISPRa prevents AMPH CPP in both males and females, as well as the increase in PFC dopamine following MPH administration in male mice. In general, this is a clear, well-written and timely manuscript. There are a requests for clarification outlined below that if addressed would make the manuscript stronger.

1. Please add postnatal day tested for adolescence and surgeries in the abstract and methodology to define early, late, or middle adolescence vs adulthood in addition to them already presented in the results.
2. Figure 1 it seems more animals are in groups from later stereology experiments that are derived / depend on the CPP data using amphetamine / saline. As one example, how is there more n per some groups in stereology female saline than in the CPP that was used as the pretreatment for the stereology groups. Please clarify.
3. Perhaps related to #2, clarification of animal N per group is needed throughout the manuscript. Why is there more animals in some PND88 CPP groups compared to PND32 CPP groups? Is this a within subject design across ages?
4. An optional stylistic change would be to put F-values, p-values, etc in the results section, as is standard. It took me a while to figure out where all the statistical parameters were located, although admittedly I do like the authors approach of embedding in the figure (hence why it is optional). Perhaps a quick note in the text saying that F-values are in embedded in the figures would save readers the search.

Minor / Typos:

1. Some text is subscripted in the stereology section of methods.
2. Some inconsistent formatting for Figure citations in text. (Fig. 1B) vs (Fig 1 C&D), etc.

Reviewer #3

(Remarks to the Author)

Hernandez and colleagues present a series of experiments demonstrating the neurochemical consequences of adolescent amphetamine exposure on dopaminergic projections to the prefrontal cortex (PFC). The manuscript suggests that in male mice, there is a re-routing of axons intended for the nucleus accumbens to instead innervate the prefrontal cortex. The experiments and methods are well described, and the paper adds to several other recent publications from the group (which are appropriately cited in the manuscript). The manuscript is well written, and the observations are quite novel and potentially impactful.

Major concerns:

1. My most significant concern is based in the authors' interpretation that the axons grow ectopically to the cortex during adolescence. It is more likely that the axons are already present at this late stage of development, but some additional branching and/or phenotypic change is present. Regarding phenotypic changes, it is well established that the norepinephrine transporter is primarily responsible for clearing dopamine in the PFC; mesocortical dopamine axons do not normally express very much dopamine transporter - the current manuscript shows that nicely in Figure 1C in their controls. Perhaps in the case of the adolescent amphetamine exposure, DAT is upregulated.
2. Figure 5 provides convincing functional data suggesting the the Dcc "rescue" normalizes the neurochemical phenotypes. Panel H also shows the presence of mCherry labeling in the VTA, indicative of successful expression of the CRISPRa constructs. But it is surprising that Dcc levels themselves are not provided. It would be especially helpful to know whether Dcc is present in the PFC dopaminergic terminals.

Lesser concerns:

1. Please provide citations or controls to demonstrate specificity of the antibodies used.
2. The letter labels in figures 4 and 5 are too small.

Version 1:

Reviewer comments:

Reviewer #1

(Remarks to the Author)

The authors have responded comprehensively to the reviewers' critiques and the manuscript is now acceptable with the following corrections of some minor contradictions in the rebuttal and text.

Comment 2 in rebuttal states that there "ARE changes in VMAT2 levels in the PFC between adult mice exposed to amphetamine in adolescence versus those exposed to saline in adolescence". This is NOT accurate; there were no changes found (only needs correction if rebuttal is published).

SFig1 legend states that tissue was taken at PND101+/-15) whereas new text in Methods says tissue was taken at PND75.

This text is also in Rebuttal.

Reviewer #2

(Remarks to the Author)

The authors did a good job responding to comments and revising manuscript. I have no additional comments that need to be addressed.

Reviewer #3

(Remarks to the Author)

The authors have revised and significantly improved their manuscript. The majority of my concerns have been addressed, and the remaining items do not preclude suitability for publication.

Reviewers' comments:

Reviewer #1 (Remarks to the Author):

In this study, the authors investigate whether adolescent amphetamine (AMPH) exposure for 12 days (PND21-32) impacts dopamine (DA) dynamics and the morphological and functional phenotype of DA axons in the medial prefrontal cortex (PFC) of young adult mice. This study builds on the fascinating body of work from this group adolescent demonstrating decreased expression of DCC and increased dopamine transporter (DAT) -expressing neurons in the medial prefrontal cortex (PFC) of male, but not female, mice. In the current study, they also discovered that there is a significantly greater expression of DAT+ axons in control male vs. female mice. After adolescent AMPH, PFC DA axons display abnormal DA transients, faster DA release, and exacerbated responses to the DAT inhibitor, methylphenidate, in young adulthood. Further, CRISPRa-mediated upregulation of the axon-guiding Netrin-1 receptor, downregulated in colorectal cancer (DCC), in adolescence prevents these changes. They conclude that ectopic PFC DA neurons in PFC rerouted from terminating in the nucleus accumbens (NAc) in adolescence retain the anatomical and functional phenotypes of NAc axons. The study is novel, appears to be well-performed, and the methods and data are well-described/illustrated, particularly the detailed statistical analysis section, but the discussion and conclusions regarding DAT expression as the primary mechanism underlying these changes are somewhat speculative and lack experimental support, weakening the study's overall impact.

1. DA reuptake/clearance was not significantly altered in adolescent male mice. The data using GRABDA2H which has a slow signal decay time did not detect a change in DA clearance in AMPH-treated male mice. A traditional method to measure clearance should be used to support their conclusion that increased DAT expression underlies the change in DA dynamics.

We appreciate the reviewer's insightful comment regarding the limitations of the GRAB_{DA2H} sensor, particularly its slow signal decay time, which may mask subtle changes in DA clearance. We agree that a faster in-vivo method would provide valuable complementary data. However, using a traditional technique like Fast-Scan Cyclic Voltammetry (FSCV) in the prefrontal cortex (PFC) presents significant challenges. The close proximity and co-release of norepinephrine make it difficult to definitively isolate and quantify DA clearance using FSCV in this brain region given that the voltammogram of dopamine and norepinephrine are nearly identical ¹

Regarding our conclusion that DAT expression is increased in adult male mice treated exposed to amphetamine in adolescence, we would like to emphasize that this conclusion is based on the results from the neuroanatomical experiment and quantitative stereological analysis showing a significant increase in the number of DAT-immunopositive varicosities in the prefrontal cortex of amphetamine pre-treated groups compared to saline-pretreated mice. Although we do not include a direct measure of DA reuptake/clearance, the changes in PFC DA dynamics we report at baseline and in response to acute methylphenidate in amphetamine pre-treated mice versus saline pre-treated groups are consistent with increased DAT function.

Our results are also in line with a growing body of evidence showing that increased DAT expression in the striatum of mice results in altered DA dynamics. Studies using a variety of techniques, including FSCV and microdialysis, have shown that changes DAT expression leads to increase in dopamine release and

content in other brain areas^{2,3}. Noticeably, DAT overexpression in mice has been shown to increase methylphenidate potency^{3,4}, aligning with our observed changes in DA signaling.

To clarify this important point, we modified the information pertaining this issue in the Discussion section (please see page 12, lines 437-449):

“Dopamine function in the PFC is highly sensitive to stress^{69,70}; we were able to measure group differences in stress-induced changes in dopamine transients via an acute i.p. injection of saline. The steeper and faster dopamine release observed in response to saline in AMPH-pretreated male mice, compared to saline counterparts, is consistent with these mice having greater PFC DAT expression and, therefore, increased dopamine reserve pools. The fact that we did not capture differences between groups in stress-induced dopamine clearance is likely due to the slow signal decay of the GrabDA2h sensor, which cannot detect differences in fast dopamine clearance events^{48,49}. While a faster technique like Fast-Scan Cyclic Voltammetry could potentially capture these dynamics, its use in the PFC is complicated by the difficulty of distinguishing dopamine signals from those of co-released norepinephrine⁷¹. However, the stereological data showing increased DAT expression in the PFC of adult male mice exposed to AMPH in adolescence, compared to their saline counterparts, is in line with the accelerated dopamine release observed in this group, manifested as a steeper slope (k ascending) and a shorter time needed for the fluorescence signal to double (doubling time)”.

2. Why wasn't VMAT2 measured to explore whether DA release changes emerge from changes in vesicular release?

Thank you for this suggestion. We now measured VMAT2 expression in the PFC of adult male mice using western blot. There are difference in VMAT2 levels in the PFC between adult mice exposed to amphetamine in adolescence versus those exposed to saline in adolescence.

i. We added this result as a supplementary Figure 1.

Supplementary Figure 1. AMPH in adolescence does not alter VMAT2 levels in the adult PFC of male mice. **A.** Experimental timeline. **B.** Male mice exposed to AMPH (4.0 mg/kg) in early adolescence developed place preference for the side of the box paired with the drug. **C.** Western blot image from adult male mice (PND101±15) exposed to saline or AMPH in adolescence (PND21). The two isoforms for vesicular monoamine transporter 2 (VMAT-2; 56kDa and 70kDa) and GAPDH reference band (~37kDa) are indicated. **D.** There is no difference in the expression of VMAT-2 in the PFC of mice that were exposed to AMPH in adolescence compared to their saline counterparts. All bar graphs are presented as mean values ±SEM. Saline n=7; Amphetamine n= 5. Detailed statistics are provided on the statistical table. Source data are provided as a Source Data file.

ii. We added the description of this experiment to the Methods section (page 5-6, line 122-150)

“For Western blot analysis, we prepared protein lysates of the prefrontal cortex of adult male mice (PND 101±15) exposed to AMPH or saline in early adolescence (PND 21-32) using the CPP paradigm. Mice were left undisturbed in their home cages and at PND75 their brains were extracted and flash frozen. Bilateral tissue punches of the PFC were collected as before¹¹. We used a RIPA lysis buffer (Milipore-Sigma) containing protease and phosphatase inhibitors (Roche) to prevent protein degradation and dephosphorylation. Protein concentrations were determined using a BCA protein assay

kit (Thermo Fisher Scientific), and we normalized samples to ensure equal protein loading. Protein lysates (20 µg) were denatured by heating at 95 °C for 5 minutes in a Laemmli loading buffer (Bio-Rad) containing β-mercaptoethanol(Sigma) We then separated the proteins by SDS-PAGE (Sodium dodecyl sulfate–polyacrylamide gel electrophoresis) on a 4–15% Mini-PROTEAN TGX Stain-Free Gels (Bio-Rad) at 180 V for 30minutes. Following electrophoresis, we transferred the separated proteins to a PVDF (polyvinylidene difluoride) membrane (Bio-Rad) using a Trans-Blot Turbo transfer system (Bio-Rad, Hercules, CA, USA) at 2.5 A and 25 V for 7 minutes.

Immunoblotting and Detection: After protein transfer, we blocked the PVDF membrane in a 5% BSA (Millipore-Sigma) in TBST (Tris-buffered saline with 0.1% Tween 20) for 1 hour at room temperature to prevent non-specific antibody binding. The membrane was then incubated overnight at 4 °C with polyclonal rabbit anti-VMAT2 (1:2500; kindly provided by Dr. G.W. Miller³³, which was diluted in the blocking solution, and with monoclonal rabbit anti-GAPDH (1:10000 , Cell signaling, 36835).

Following primary antibody incubation, we washed the membrane 3 times for 10 minutes each in TBST. We then incubated the membrane with Goat Anti-Rabbit IgG Antibody (H+L), Biotinylated (1:5000; vector laboratories, BA-1000-1.5) and streptavidin-HRP (1:5000; Thermofisher, 434323) each for 1 hour at room temperature. After three more washes in TBST, we detected the protein bands using a Clarity chemiluminescence (ECL) Western blotting substrate (Biorad) and visualized them using a ChemiDoc XRS+ imaging system (Bio-Rad). We used densitometry to quantify the band intensities using Imagelab (Biorad), and we normalized the protein expression levels to GAPDH, to ensure accurate comparisons between samples. One of the samples obtained from the AMPH treated group was removed from the analysis since it was an outlier.”

iii. We added the following to the Results section (page 8, line 271-281)

“AMPH in adolescence does not increase vesicular monoamine transporter 2 (VMAT2) expression in the adult PFC of male mice

In addition to the reuptake of extracellular dopamine by DAT, dopamine is accumulated and compartmentalized in cytoplasmic vesicles by VMAT2, thereby regulating dopamine concentrations within neuronal compartments^{43,44}. To verify if AMPH in adolescence alters the expression of VMAT2 in the adult PFC of male mice, we used western immunoblot of PFC tissue of adult males mice exposed to AMPH or saline in

adolescence. The VMAT2 antibody detected two bands at approximately 70, 56 kDa, in agreement with previously reported molecular weights for this protein following post-translational modifications in mice^{33,45}. The levels of adult VMAT2 isoforms in the PFC do not differ between AMPH and saline treated groups (Supplementary Fig. 1). “

iv. We added the following paragraph to the Discussion section (page 12, lines 428-435)

“There is some evidence suggesting that DA axons in the NAc and in the PFC also have different levels of the vesicular transporter VMAT2^{22,67}, we therefore assessed if the impact of AMPH in adolescence on DAT expression in the adult PFC in male mice would also extend to changes in VMAT2 levels. We did not observe changes in VMAT2 PFC levels in AMPH-treated mice, likely due to the limitation of the Western blot analysis to detect subtle changes in protein levels within axon terminals. Another potential factor is the expression of VMAT2 in astrocytes within the PFC⁶⁸, which may mask any potential changes occurring in dopamine terminals. Future experiments should include stereological quantification of VMAT2+ axons to shed light into this issue.”

3. The authors should investigate whether adolescent methylphenidate has the same effect as AMPH (an experiment not required for this study).

Thank you for this suggestion. Indeed we are very interested in comparing effects of AMPH versus methylphenidate exposure in adolescence on dopamine development. Our interest stems from their differential mechanism by which they alter dopamine function and on intriguing and compelling evidence in humans showing that detrimental behavioral effects of AMPH and methylphenidate can be quite different, with AMPH but not methylphenidate exposure being associated with increased odds (greater risk) of psychosis or mania, in adolescents and young adults^{5,6}. In addition, we explored the effects of methylphenidate exposure on cocaine reward and DCC receptor expression in the VTA in adulthood and found changes suggestive of protective processes⁷. We are eager to continue pursuing this line of work.

4. Without data from Points 1 and 2, the discussion remains speculative and the conclusions weakened.

We added data and expanded the Discussion section addressing the issues that were raised in points 1 and 2. We believe that these additions strengthen the conclusions stated.

Minor Points

1. Introduction-The first paragraph cites studies done in a variety of unidentified species. Please identify the different species in the literature cited.

We have indicated the species in which the studies were performed across the Introduction.

2. When referring to AMPH doses in rodents throughout the manuscript, please refer to rewarding but not “recreational” doses that are only relevant to humans.

Thank you for this comment. We prefer to continue referring to the AMPH regimen we used as “recreational-like” to be consistent with all our previous work and because we have shown that this dose achieves peak plasma levels in adolescent mice comparable to those seen in human recreational use⁸. We now make sure to always indicate “recreational-like” regimen. We also indicate across the manuscript that this dose regimen is rewarding.

3. Second sentence of second paragraph to improve clarity: Which region are you referring to here “...reduced number of varicosities in this region”? Which region-PFC or NAc?

To improve clarity, we changed the word “region” for “PFC”:

“These disruptions cause mistargeting of dopamine axons, leading to the rerouting of mesolimbic projections to the PFC and to a reduced number of dopamine axon varicosities in this region^{9,21,22}.”

Reviewer #2 (Remarks to the Author):

The manuscript by Hernandez and colleagues describes an interesting study showing that repeated amphetamine administration, which induces CPP in male and females, leads to an increased expression of prefrontal cortex (PFC) DAT in male, but not female mice. There is also a corresponding increase in dopamine amplitude and transients at baseline, and increased amplitude following saline and methylphenidate administration in male, but not female, PFC. Finally, amplification of DCC expression via CRISPRa prevents AMPH CPP in both males and females, as well as the increase in PFC dopamine following MPH administration in male mice. In general, this is a clear, well-written and timely manuscript. There are a request for clarification outlined below that if addressed would make the manuscript stronger.

1. Please add postnatal day tested for adolescence and surgeries in the abstract and methodology to define early, late, or middle adolescence vs adulthood in addition to them already presented in the results.

The postnatal day was added to the Abstract and in Methods section.

2. Figure 1 it seems more animals are in groups from later stereology experiments that are derived / depend on the CPP data using amphetamine / saline. As one example, how is there more n per some groups in stereology female saline than in the CPP that was used as the pretreatment for the stereology groups. Please clarify.

Two male subjects who received AMPH were excluded from the stereological analysis because their coronal brain sections were damaged during processing and handling for immunofluorescence, making them unsuitable for stereological analysis.

To explain this point, we made the following changes:

i. We now matched the groups and re-analyzed the place preference data after excluding the males whose brain sections were not available for the stereology experiment. The figures and statistical tables have been updated accordingly. Please see below the CPP data.

ii. We now include the following in the Methods section (Page 6 lines 173-175):

“Two male subjects who received AMPH were excluded from the stereological analysis because their coronal brain sections were damaged during processing and handling for immunofluorescence, making them unsuitable for stereological analysis.”

3. Perhaps related to #2, clarification of animal N per group is needed throughout the manuscript. Why is there more animals in some PND88 CPP groups compared to PND32 CPP groups? Is this a within subject design across ages?

Thank you for pointing at this issue.

i. There is no difference in the sample size of the data shown for the CPP test performed at PND88 compared to PND32 CPP groups. To avoid potential confusion, we included the sample size (n) for each group in the figure captions and added the following statement at the end of each caption:

“Comprehensive statistics are available in the Statistics Table, while the underlying source data for the graphs is presented in the corresponding source table”

ii. We now explain why there is discrepancy in the sample size in this experiment in the Methods section (page 4 lines, 66-67; 80-89):

“During the fiber photometry experiment, one male mouse from the saline group and one male mouse from the AMPH group, both exposed during adolescence, were excluded due to insufficient signal. Among the remaining subjects, an additional male mouse from the adolescent saline group did not exhibit a distinct saline peak and was therefore omitted from this portion of the analysis. For female mice, data could not be obtained from two individuals treated with AMPH and two treated with saline in adolescence, owing to either lack of signal or loss of the headcap containing the optical fiber. Regarding animals subjected to CRISPRa, two mice from the LacZ group, one which received saline and another one which received AMPH during adolescence did not show fluorescent signal. One individual from the Dcc overexpression group, which received AMPH during adolescence, was excluded as an outlier from the analysis.”

4. An optional stylistic change would be to put F-values, p-values, etc in the results section, as is standard. It took me a while to figure out where all the statistical parameters were located, although admittedly I do like the authors approach of embedding in the figure (hence why it is optional). Perhaps a quick note in the text saying that F-values are embedded in the figures would save readers the search.

We are now including a separate detailed statistics table in all our publications because we find that this table provides an easy, consistent and systematic way of describing all the statistical results and parameters measured. To make this clear, we added the following sentence to the Methods section (page 8, lines 240-245)

“Descriptive statistics can be found in the descriptive statistics table. The underlying source data for the graphs is presented in the source table. All statistical tests performed are described in Tables 1–5. When significant, the results of the statistical analysis are embedded in the figures. Statistical significance was defined as $p \leq 0.05$. Normality was assessed using Shapiro-Wilk tests. Experimenters were blinded to group allocation during neuroanatomical analysis. All statistical analyses were carried out using Prism software (GraphPad), except for the GEE analysis, which was done in SPSS”

Minor / Typos:

1. Some text is subscripted in the stereology section of methods.

We went over this section and made sure no text is subscripted, except for GRAB_{DA}.

2. Some inconsistent formatting for Figure citations in text. (Fig. 1B) vs (Fig 1 C&D), etc.

The inconsistent formatting was resolved.

Reviewer #3 (Remarks to the Author):

Hernandez and colleagues present a series of experiments demonstrating the neurochemical consequences of adolescent amphetamine exposure on dopaminergic projections to the prefrontal cortex (PFC). The manuscript suggests that in male mice, there is a re-routing of axons intended for the nucleus accumbens to instead innervate the prefrontal cortex. The experiments and methods are well described, and the paper adds to several other recent publications from the group (which are appropriately cited in the manuscript). The manuscript is well written, and the observations are quite novel and potentially impactful.

Major concerns:

1. My most significant concern is based in the authors' interpretation that the axons grow ectopically to the cortex during adolescence. It is more likely that the axons are already present at this late stage of development, but some additional branching and/or phenotypic change is present.

i. Our laboratory has conducted comprehensive research on this topic, as detailed in Reynolds et al. (2018), where we directly tracked the growth of dopamine axons in adolescence. Employing an axon-initiated recombination technique, a Cre recombinase (CAV-Cre) virus was injected into the nucleus accumbens (NAcc), while a Cre-dependent virus expressing enhanced yellow fluorescent protein (eYFP) (pAAV-Ef1a-DIO-EYFP-WPRE-pA), was simultaneously microinjected into the ventral tegmental area (VTA). This methodology ensures that only those VTA neurons with axons extending to the NAc by postnatal day 21 are labeled with eYFP. In our study, we identified eYFP-positive dopamine axons in the prefrontal cortex (PFC) of adult mice that received these injections during adolescence.

To exclude the possibility that these eYFP-positive dopamine axons are collaterals of fibers innervating the NAc, we performed the same axon-initiated viral tracing experiment in adult wild-type mice. We found that eYFP-positive dopamine axons in the cingulate, infralimbic and prelimbic subregions of the PFC are absent or negligible in mice that received viral injections in adulthood and were euthanized 6 weeks later. This is in line with reports showing that 1) the vast majority of dopamine axons do not send collaterals between the NAc and the PFC⁹⁻¹⁴, and 2) dopamine axon growth into the PFC is complete by PND 60 in rodents^{15,16}.

ii. In the same study by Reynolds et al., (2018), we quantify the complexity of individual dopamine axons in the PFC by adapting the axonal complexity index from developmental studies using *Xenopus* tadpoles. For this experiment we used a Cre-dependent fluorophore virus DIO-eYFP into the VTA. Four to 5 weeks later, only eYFP+ axons with intact arbors, defined as having all of the tips of their terminal branches within the section, were included in the analyses. We then used NeuroLucida software to quantify axon arbor length, branch order, and varicosity density of each axon. We found that adolescent dopamine axon growth in the PFC is not associated with increased branching of axons that have innervated this region at an earlier time. While AMPH in adolescence induces ectopic growth of NAc axons to the PFC, it does not alter dopamine axon branching complexity¹⁷.

iii. We modified the following sentence In the Discussion section (page 13, lines 503-506):

“Increasing Dcc gene expression in the VTA of male mice during adolescence, using CRISPRa, prevented the exaggerated dopamine response to methylphenidate in adulthood. This supports the idea that the mesolimbic-like dopamine phenotype in the PFC is caused by AMPH-induced rerouting of NAc dopamine axons to the PFC and that changes in dopamine axon branching are not involved”^{11,13,18}.

Regarding phenotypic changes, it is well established that the norepinephrine transporter is primarily responsible for clearing dopamine in the PFC; mesocortical dopamine axons do not normally express very much dopamine transporter - the current manuscript shows that nicely in Figure 1C in their controls. Perhaps in the case of the adolescent amphetamine exposure, DAT is upregulated.

What our results show is exactly that: increased DAT+ expression in the adult PFC of male mice exposed to AMPH in adolescence compared to saline controls. As we mentioned across the manuscript, this NAc-like phenotype most likely results from the ectopic growth of NAc dopamine axons to the PFC, as we showed in our previous study¹⁸.

In the NAc, all dopamine axons express DCC receptors¹⁹. In contrast, in the PFC, dopamine axons express negligible or zero levels of DCC¹⁹. Notably, reducing DCC expression in NAc dopamine axons in adolescence triggers their ectopic growth to the PFC. These ectopic axons express DCC receptors in the adult PFC²⁰.

We now performed an immunofluorescence experiment on coronal PFC sections of an adult male mouse receiving the intersectional viral infection in early adolescence to label dopamine axons growing from the NAc to the PFC. We processed the brain sections for DCC and DAT immunofluorescence and found, as expected, the presence of DCC+/DAT+ axons in adult male mice exposed to AMPH in early adolescence but not in their saline counterparts. This is consistent with the idea that ectopic NAc dopamine axons in the PFC preserve phenotypic characteristics of their intended target.

We also observe neurons expressing DCC in the adult PFC, consistent with our previous characterization of DCC receptor expression in this region¹⁹. Please see photomicrograph below showing this finding.

2. Figure 5 provides convincing functional data suggesting the the *Dcc* "rescue" normalizes the neurochemical phenotypes. Panel H also shows the presence of mCherry labeling in the VTA, indicative of successful expression of the CRISPRa constructs. But it is surprising that *Dcc* levels themselves are not provided.

This is a very valid point. We needed to run a separate experiment to quantify *Dcc* levels using qPCR. However, in our previous study¹⁸ we performed extensive validation experiments of the CRISPRa manipulation and demonstrated that there is a correlation between the increase in *Dcc* mRNA in the VTA and DCC protein expression in the NAc¹⁸. Since performing this separate experiment will require a significant amount of time, we are confident that the immunofluorescent findings shown in Figure 5- panel H, together with the prior robust characterization of this gene editing approach, would strengthen the results we show in this part of the study.

It would be especially helpful to know whether *Dcc* is present in the PFC dopaminergic terminals.

Please see the second part of our reply to point 1.

Lesser concerns:

1. Please provide citations or controls to demonstrate specificity of the antibodies used.

Detailed information about the antibodies used in the experiments is provided in the reporting summary as required by the journal. We now also added the citations to the studies in which these antibodies have been used and validated.

Antibodies used

Polyclonal rabbit anti TH antibody AB152

<https://www.sigmaaldrich.com/CA/en/product/mm/ab152>

Monoclonal mouse anti TH antibody (Millipore Sigma Cat No. MAB 318)

<https://www.sigmaaldrich.com/CA/en/product/mm/mab318>

Monoclonal rat anti-dopamine transporter (DAT) antibody (Millipore sigma MAB369)

<https://www.sigmaaldrich.com/CA/en/product/mm/mab369>

Polyclonal rabbit anti-VMAT2

<https://pubmed.ncbi.nlm.nih.gov/27836486/>

Rockland Rabbit antiRFP

<https://www.rockland.com/categories/primary-antibodies/rfp-antibody-pre-adsorbed-600-401-379/?srsltid=AfmBOorj9MjZ-BAzKcPK0A8LulqZdfDDz3o331Vrl89G-5gRzbSVYQ>

AF conjugated secondaries

<https://www.thermofisher.com/ca/en/home/life-science/antibodies/secondary-antibodies/fluorescent-secondary-antibodies/alexa-fluor-secondary-antibodies.html>

Cliburn, R. A. *et al.* Immunochemical localization of vesicular monoamine transporter 2 (VMAT2) in mouse brain. *J. Chem. Neuroanat.* **83**, 82–90 (2017).

Reynolds, L. M. *et al.* Dopaminergic System Function and Dysfunction: Experimental Approaches. *Neuromethods* 31–63 (2022) doi:10.1007/978-1-0716-2799-0_2.

Miller, G. W. *et al.* Immunochemical analysis of dopamine transporter protein in Parkinson's disease. *Ann. Neurol.* **41**, 530–539 (1997).

Nasirova, N., Quina, L. A., Novik, S. & Turner, E. E. Genetically Targeted Connectivity Tracing Excludes Dopaminergic Inputs to the Interpeduncular Nucleus from the Ventral Tegmentum and Substantia Nigra. *eNeuro* **8**, ENEURO.0127-21.2021 (2021).

2. The letter labels in figures 4 and 5 are too small.

Thank you for noticing this. We made the fonts larger for all the figures.

References:

1. Robinson, D. L., Venton, B. J., Heien, M. L. & Wightman, R. M. Detecting Subsecond Dopamine Release with Fast-Scan Cyclic Voltammetry in Vivo. *Clin. Chem.* **49**, 1763–1773 (2003).

2. Jones, S. R. *et al.* Profound neuronal plasticity in response to inactivation of the dopamine transporter. *Proc. Natl. Acad. Sci.* **95**, 4029–4034 (1998).

3. Calipari, E. S., Ferris, M. J., Siciliano, C. A. & Jones, S. R. Differential Influence of Dopamine Transport Rate on the Potencies of Cocaine, Amphetamine, and Methylphenidate. *ACS Chemical Neuroscience* **6**, 155–162 (2015).

4. Calipari, E. S., Ferris, M. J., Salahpour, A., Caron, M. G. & Jones, S. R. Methylphenidate amplifies the potency and reinforcing effects of amphetamines by increasing dopamine transporter expression. *Nat. Commun.* **4**, 2720 (2013).

5. Moran, L. V. *et al.* Psychosis with Methylphenidate or Amphetamine in Patients with ADHD. *N. Engl. J. Med.* **380**, 1128–1138 (2019).

6. Moran, L. V. *et al.* Risk of Incident Psychosis and Mania With Prescription Amphetamines. *Am. J. Psychiatry* **181**, 901–909 (2024).

7. Argento, J. K., Arvanitogiannis, A. & Flores, C. Juvenile exposure to methylphenidate reduces cocaine reward and alters netrin-1 receptor expression in adulthood. *Behav. Brain Res.* **229**, 202–207 (2012).

8. Cuesta, S. *et al.* DCC-related developmental effects of abused- versus therapeutic-like amphetamine doses in adolescence. *Addict. Biol.* **25**, e12791 (2020).
9. Björklund, A. & Dunnett, S. B. Dopamine neuron systems in the brain: an update. *Trends Neurosci.* **30**, 194–202 (2007).
10. Beier, K. T. *et al.* Circuit Architecture of VTA Dopamine Neurons Revealed by Systematic Input-Output Mapping. *Cell* **162**, 622–634 (2015).
11. Aransay, A., Rodríguez-López, C., García-Amado, M., Clascá, F. & Prensa, L. Long-range projection neurons of the mouse ventral tegmental area: a single-cell axon tracing analysis. *Front. Neuroanat.* **9**, 59 (2015).
12. Loughlin, S. E. & Fallon, J. H. Substantia nigra and ventral tegmental area projections to cortex: Topography and collateralization. *Neuroscience* **11**, 425–435 (1984).
13. Yetnikoff, L., Lavezzi, H. N., Reichard, R. A. & Zahm, D. S. An update on the connections of the ventral mesencephalic dopaminergic complex. *Neuroscience* **282**, 23–48 (2014).
14. Lammel, S. *et al.* Unique Properties of Mesoprefrontal Neurons within a Dual Mesocorticolimbic Dopamine System. *Neuron* **57**, 760–773 (2008).
15. Kalsbeek, A., Voorn, P., Buijs, R. M., Pool, C. W. & Uylings, H. B. M. Development of the dopaminergic innervation in the prefrontal cortex of the rat. *J. Comp. Neurol.* **269**, 58–72 (1988).
16. Naneix, F., Marchand, A. R., Scala, G. D., Pape, J.-R. & Coutureau, E. Parallel maturation of goal-directed behavior and dopaminergic systems during adolescence. *J. Neurosci.* **32**, 16223–32 (2012).
17. Reynolds, L. M. *et al.* Early Adolescence is a Critical Period for the Maturation of Inhibitory Behavior. *Cerebral cortex (New York, N.Y. : 1991)* 1–11 (2018) doi:10.1093/cercor/bhy247.
18. Reynolds, L. M. *et al.* Amphetamine disrupts dopamine axon growth in adolescence by a sex-specific mechanism in mice. *Nat Commun* **14**, 4035 (2023).
19. Manitt, C. *et al.* The netrin receptor DCC is required in the pubertal organization of mesocortical dopamine circuitry. *The Journal of neuroscience : the official journal of the Society for Neuroscience* **31**, 8381–94 (2011).
20. Reynolds, L. M. *et al.* DCC Receptors Drive Prefrontal Cortex Maturation by Determining Dopamine Axon Targeting in Adolescence. *Biol. Psychiatry* **83**, 181–192 (2018).